# Molecular Genetic Epidemiology of an Emerging Antimicrobial-Resistant *Klebsiella pneumoniae* Clone (ST307) Obtained from Clinical Isolates in Central Panama

**DOI:** 10.3390/antibiotics11121817

**Published:** 2022-12-14

**Authors:** Virginia Núñez-Samudio, Gumercindo Pimentel-Peralta, Mellissa Herrera, Maydelin Pecchio, Johana Quintero, Iván Landires

**Affiliations:** 1Instituto de Ciencias Médicas, Las Tablas, Los Santos 0710, Panama; 2Sección de Epidemiología, Departamento de Salud Pública, Región de Salud de Herrera, Ministry of Health, Chitré, Herrera 0601, Panama; 3Laboratorio Clínico, Hospital Luis Chicho Fábrega, Región de Salud Veraguas, Ministry of Health, Santiago, Veraguas 0923, Panama; 4Unidad de Infectología, Hospital Dr. Gustavo N. Collado R, Caja de Seguro Social, Chitré, Herrera 0601, Panama; 5Hospital Regional Dr. Joaquín Pablo Franco Sayas, Región de Salud de Los Santos, Ministry of Health, Las Tablas, Los Santos 0710, Panama

**Keywords:** epidemiology, molecular genetics, antimicrobial resistance, *Klebsiella pneumoniae*, emerging clone ST307, Panama

## Abstract

*Klebsiella pneumoniae* has been among the main pathogens contributing to the burden of antimicrobial resistance (AMR) in the last decade, and *K. pneumoniae* AMR strains predominantly cluster in the ST258 clonal complex. However, ST307 is emerging as an important high-risk clone. In Central America, there have been few studies on the molecular epidemiology of the *K. pneumoniae* strains involved in infections. Materials and Methods: We conducted an epidemiological study in three reference hospitals in the central region of Panama, using isolates of *K. pneumoniae* involved in infections, and identifying their AMR profile, associated clinical risk factors, and molecular typing using a multilocus sequence typing (ST) scheme. Results: Six STs were detected: 307 (55%), 152, 18, 29, 405, and 207. CTX-M-15- and TEM-type beta-lactamases were identified in 100% of ESBL-producing strains; substitutions in *gyrA* Ser83Ile and parC Ser80Ile were identified in all ST307s; and in ST152 gyrA Ser83Phe, Asp87Ala, and parC Ser80Ile, the *qnrB* gene was detected in all strains resistant to ciprofloxacin. Conclusions: We present the first report on ST307 in three reference hospitals in the central region of Panama, which is a high-risk emerging clone and represents a public health alert for potential difficulties in managing *K. pneumoniae* infections in Panama, and which may extend to other Central American countries.

## 1. Introduction

*Klebsiella pneumoniae* is an opportunistic enterobacterium that is responsible for infections in susceptible populations, such as the elderly, neonates, and patients with diabetes or immunosuppressed states, as well as healthcare-associated infections [1]. *K. pneumoniae* represents one of the main pathogens contributing to the burden of antimicrobial resistance (AMR) and associated deaths [2], which is why the World Health Organization (WHO) includes *K. pneumoniae* with AMR in the list of Priority 1 (critical group) pathogens resistant to antibiotics [3].

β-lactam and fluoroquinolone antibiotics are widely prescribed to treat infections caused by *K. pneumoniae* [4]. However, AMR to this antibiotic group is increasing worldwide [5]. *K. pneumoniae* can undergo various mechanisms conferring resistance to commonly used antibiotics, the most frequent of which is extended-spectrum β-lactamases (ESBL) such as SHV, TEM, and CTX-M: a group of enzymes that confer resistance to oxyminocephalosporins (i.e., cefotaxime, ceftazidime, cefuroxime, cefepime) and to monobactams (i.e., aztreonam), but not cephamycins (i.e., cefoxitin, cefotetan) or carbapenems (i.e., imipenem, meropenem, ertapenem) [6,7,8]. Within the mechanisms of resistance to quinolones, *K. pneumoniae* may contain chromosomal amino acid substitutions within the regions determining resistance to quinolones (QRDR) of the *gyrA* and *parC* genes, which are the targets of quinolones. A third resistance mechanism described for *K. pneumoniae* is the acquisition of plasmid-mediated quinolone resistance (PMQR) genes, among which the *qnr* genes (*qnrA*, *qnrS*, *qnrB*, *qnrC*, and *qnrD*) and aac(6’)-Ib-cr have typically been identified [9].

*K. pneumoniae* strains with AMR mainly belong to certain sequence types (STs) that represent high-risk international clones. In the last decade, hospital outbreaks have been predominantly attributed to isolates belonging to the clonal complex 258 (i.e., ST258, ST11, and ST512) [10]. However, ST307 has emerged in different parts of the world, with involvement in hospital outbreaks in Africa, Asia, Europe, and the Americas [11]. In Latin America, ST307 has been described in the last decade in countries such as Colombia [12], Brazil [13,14], Mexico [15], and Ecuador [16].

*K. pneumoniae* ST307 is frequently associated with AMR, especially because it tends to carry plasmid-mediated ESBL CTX-M-15 and carbapenemases, which hydrolyze carbapenems. These plasmids also carry genes conferring resistance to aminoglycosides and quinolones, emerging globally as an important AMR organism [4]. The most commonly identified carbapenemases in ST307 are KPC-2, KPC-3, OXA-48, and NDM-1. In addition, resistance to combined beta-lactam inhibitors, such as ceftazidime/avibactam and colistin, has been reported [17]. Previous studies have identified chromosome- and plasmid-encoded mechanisms of colistin resistance [18,19]. Currently, in Central America, knowledge of the epidemiological and molecular characteristics of the circulating *K. pneumoniae* strains remains limited. A study conducted on *K. pneumoniae* isolates carrying blaKPC carbapenemase collected between 2006 and 2015 in various Central American countries, including Panama, identified the high-risk clone ST258 [20].

It is known that a better understanding of the epidemiology of circulating *K. pneumoniae* strains is crucial to identify the factors contributing to the spread of resistance genes and is essential for the development of specific strategies for infection prevention, control, and new therapeutic strategies [21]. The purpose of this study was to characterize, through molecular epidemiology, the strains of *K. pneumoniae* isolated in the clinical laboratories of hospitals in the central region of Panama during 2018–2019 and involved in infections in patients prior to the COVID-19 pandemic.

## 2. Materials and Methods

### 2.1. Study Design

We conducted a prospective epidemiological study between October 2018 and November 2019 in three reference hospitals in central Panama: Hospitals A and B located in the Azuero region, and Hospital C located in the province of Veraguas. The three hospitals represent the main centers providing medical and laboratory care in central Panama.

### 2.2. K. pneumoniae Isolates

During the study period, we included *K. pneumoniae* samples that (a) were isolated from various outpatient and inpatient samples within the first 48 h of admission, (b) were collected as part of routine patient care procedures, and (c) showed resistance to at least one of the antibiotics routinely tested in hospital clinical microbiology laboratories. In vitro antimicrobial activity was determined using the Vitek2 system (BioMérieux; Marcy l’Etoile, France). The test results were interpreted according to the breakpoints defined by the Clinical Laboratory Standards Institute (CLSI) [22].

A technical sheet was completed anonymously for each sample collected, recording the following risk factors: age, sex, hospitalization for 2 or more days in the previous 90 days, antibiotic treatment in the previous 90 days, personal history of immunosuppressive therapy, home wound care, hemodialysis within the previous 90 days, and outpatient chemotherapy.

### 2.3. Statistical Analyses

Data was recorded in MS Excel (The Microsoft Corporation; Redmond, WA, USA). Data analyses were conducted in Stata v. 11.0 (StataCorp, LLC; College Station, TX, USA). We calculated descriptive statistics and estimates with their respective 95% confidence intervals (CIs). We used Fisher’s exact test to compare proportions, and Mann–Whitney’s U test to compare medians, setting alpha to 0.05 for statistical significance.

### 2.4. Molecular Typing Analysis and Molecular Identification of β-Lactamase

Molecular typing analyses were conducted using Pasteur’s multilocus sequence typing (MLST) scheme. We performed MLST schemes using a standardized protocol specific for *K. pneumoniae* [23]. Internal sequencing fragments of seven internal genes (i.e., *rpoB*, *gapA*, *mdh*, *pgi*, *phoE*, *infB*, and *tonB*) were amplified from the chromosomal DNA of the *K. pneumoniae* strains. Sequencing of polymerase chain reaction (PCR) amplicons was performed using the services of Macrogen (Macrogen Inc.; Seoul, Republic of Korea). Gene sequences were analyzed using Geneious prime v.2020.5 (Biomatters, Ltd.; Auckland, New Zealand), and allelic profiling was determined using *K. pneumoniae* MLST databases [24]. All isolates with β-lactam and quinolone resistance phenotypes were analyzed for *blaCTX-M*, *blaTEM*, *blaSHV*, *qnrA*, *qnrB*, *qnrS*, *gyrA*, and *parC* genes using specific PCR primers, as previously described. Table 1 summarizes these procedures [25,26,27,28,29]. The PCR amplicons were sequenced and analyzed in the BLASTN program of the National Center for Biotechnology Information (NCBI; https://blast.ncbi.nlm.nih.gov/Blast.cgi, accessed on 15 July 2022). Multiple alignment and assembly of *gyrA* and *parC* was performed with the partial sequences AF052258.1 and AF303641.1 obtained from the NCBI (https://www.ncbi.nlm.nih.gov/nuccore/AF052258.1, accessed on 15 July 2022 and https://www.ncbi.nlm.nih.gov/nuccore/AF303641.1, accessed on 15 July 2022).

## 3. Results

A total of 11 strains of *K. pneumoniae* with antimicrobial resistance were analyzed, 55% (6/11) from urine cultures, 27% (3/11) from secretions (one endotracheal and two wound samples), and 18% from blood cultures and rectal swabs (1/11 each). Among the samples, 90% (10/11) came from hospitalized patients and 10% (1/11) from an outpatient. Most (64%, 7/11) patients were male and 36% (4/11) were female, with a median (IQR) age of 63 (56–75) years.

Table 2 shows the phenotypic and genotypic profile of antimicrobial resistance of the *K. pneumoniae* strains analyzed. Sensitivity analyses of the strains showed that 91% (10/11) were sensitive to carbapenems (meropenem, imipenem, ertapenem), 82% (9/11) were resistant to trimetroprim-sulfomethoxazole; 73% (8/11) were resistant to gentamicin, and 64% (7/11) were resistant to ciprofloxacin. CTX-M-15 and TEM-type beta-lactamases were identified in all nine (82%) ESBL-producing strains (Table 2). Using Pasteur’s MLST technique, all 11 strains were typed and six STs were identified among them: 55% (6/11) belonged to ST307 and 45% (5/11) belonged to STs 152, 18, 29, 405, and 2073 (Table 2). Among the ESBL-carrying strains, 67% belonged to ST307, while 33% belonged to the other identified STs (152, 405, and 2073) (*p* = 0.06). All (100%, 6/6) of the ST307 strains were resistant to quinolones, while only 20% (1/5) of the strains grouped in the other STs were resistant (*p* = 0.01). Of note, 83% of ST307 strains were resistant to gentamicin, while only 40% (2/5) of strains from the other STs were resistant to gentamicin (*p* = 0.24).

Table 3 shows the identified PMQRs and QRDRs. We observed that of all the strains with resistance to quinolones, the *qnrB* gene was detected in all strains resistant to ciprofloxacin and in strain O-2723 with intermediate sensitivity. The *qnrA* gene was detected in strain O-2723 that showed resistance to nalidixic acid. QRDR analysis determined *gyrA*: Ser83Ile and parC Ser80Ile substitutions in all ST307s and in ST152 gyrA: Ser83Phe, Asp87Ala, and parC Ser80Ile.

Table 4 summarizes the risk factors identified, along with the median age and distribution by sex, grouped into strains belonging to ST307 and strains belonging to the other STs (i.e., 152, 18, 29, 405, and 2073, grouped under “other STs”).

## 4. Discussion

*K. pneumoniae* strains with AMR in the last decade were mainly grouped into certain types of high-risk international clones, predominantly belonging to the clonal complex 258 (i.e., ST258, ST11, and ST512) [11]. However, ST307 is emerging as an important AMR clone. In this study, through molecular analysis with MLST, we identified the presence of the emerging clone ST307 in 55% of *K. pneumoniae* strains isolated between 2018 and 2019 in three hospitals in the central region of Panama.

ESBL-producing strains of *K. pneumoniae* have increased in frequency and severity worldwide, causing an impact on the prolongation of hospital stays and delays in appropriate antibiotic therapy, which have led to an increase in healthcare costs [2,30]. We observed that ESBL-producing *K. pneumoniae* strains mostly belonged to ST307 (67%), identifying the blaCTX-M-15 gene in all ESBL-carrying strains. It has been reported that the movement of plasmids between different species and lineages of enterobacteria represents an important source of AMR; an example of this is the pandemic clone of *Escherichia coli* ST131, which contributes to the dissemination of ESBL genes (blaCTX- M-15) among Enterobacteriaceae [6,31]. This pandemic clone *E. coli* ST131, which is a carrier of CTX-M-15, was identified in a recent study in clinical isolates responsible for infections in outpatients and hospitalized patients in Panama, which suggests a high prevalence among Enterobacteriaceae [32].

Most of the global isolates of ST307 described in the literature carry the blaCTX-M-15 gene on FIB-like plasmids and contain several additional AMR determinants responsible for resistance to aminoglycosides, quinolones (*qnrB1*), and other antimicrobial agents [11,33,34]. These data coincide with our findings, where all ST307 strains were resistant to ciprofloxacin and 83% to gentamicin. Our genetic analyses showed that the *qnrB* gene was detected in 100% of the ST307 strains, and the QRDR substitutions were observed in gyrA Ser83Ile and parC Ser80Ile in all strains with ST307. Sequencing studies of *K. pneumoniae* ST307 [11] described that the clone ST307 emerged around 1994 and consists of two deep branching lineages. One lineage containing the gyrA Ser83Ile and parC Ser80Ile substitutions has shown a global distribution and the other lineage, also containing an additional gyrA D87N substitution, has only been present in Texas, United States. One report [11] also proposed there was genomic evidence of between-country movement of patients infected or colonized with isolates of ST307 that belonged to the global lineage. Our data showed that the substitutions observed in the QRDR of the ST307 identified in this study corresponded to the global lineage. This being the first report of clone ST307 in Panama, its origin is not clear. It is plausible that a sensitive strain has acquired a plasmid-carrying blaCTXM-15 from other Enterobacteriaceae (e.g., *E. coli* ST131), as has previously been identified [32,35], or that a strain of ST307 was introduced before 2018 from another source, which could partly explain the fact that it was distributed in the three participating hospitals, and in different hospitalization wards, such as surgery, internal medicine, the intensive care unit, and outpatient wards.

In a recent study conducted in Colombia with *K. pneumoniae* isolates, the SHV enzyme (SHV-11 or SHV-1) was identified in all strains studied, as well as other TEM enzymes such as TEM-11 and TEM-1. These data are consistent with the present study, where SHV and TEM were identified in all strains with the ESBL phenotype [12].

In Latin America, the percentage of ESBL-producing *K. pneumoniae* strains is estimated at 24.7%, a percentage that has increased in the last decade, surpassed only by the Asia-Pacific region [5]. This scenario represents a notable problem, as it demands the use of broader-spectrum antimicrobials, such as carbapenems, resistance to which has already spread worldwide. Correlations between the use of carbapenems in hospitalized patients and the development of resistance have been demonstrated, even after up to 3 months of treatment [36]. The Latin American Antimicrobial Resistance Surveillance Network (ReLAVRA) published a growing trend in *K. pneumoniae* resistant to carbapenems, with resistance rates reaching an average of 21% [37]. ST307 associates with blaCTX-M-15; however, sufficient indexed literature supports that this lineage could acquire and spread carbapenemases (blaKPC, blaNDM, blaOXA-48, blaOXA-48, blaOXA-181, and blaGES-5) [34]. In Latin America, the first ST307 described was in Colombia in 2015 in a strain of *K. pneumoniae* carrying blaKPC-2 [12], but it has also been described in Brazil [13,14], Mexico [15], and Ecuador [16]. ST307 behaves as an emerging high-risk clone, whose genetic characteristics contribute to its adaptation to the hospital environment, as well as its potential to acquire carbapenemase-carrying plasmids. The rational use of antibiotics and surveillance are essential to counteract the high potential for plasmid acquisition.

The small number of samples is a noteworthy limitation of this study. The sample size was due, first, to the infrequent request for cultures by the attending physicians and, second, to the limited human and infrastructure resources for processing cultures in clinical laboratories in central Panama.

Limitations aside, this study makes unprecedented contributions to the knowledge of the microbiology and molecular genetic epidemiology of *K. pneumoniae* strains in Panama and Central America by identifying several STs, including the emerging clone ST307. This draws our attention to the potential difficulties in the treatment of infections originating in hospitals and the community, as well as to the importance of knowing the composition and distribution of antibiotic resistance genotypes, as an important step to establish public policies aimed at limiting the impact of AMR *K. pneumoniae* infections.

## Figures and Tables

**Table 1 antibiotics-11-01817-t001:** Primers used in the study.

Target	Primer	5′-3′ Sequence	Temperature (°C)	Amplicon (bp)	Reference
*CTX-M*	CTX-M-F	ATGTGCAGYACCAGTAARGTKATGGC	55	592	25
	CTX-M-R	TGGGTRAARTARGTSACCAGAAYSAGCGG			
*TEM*	TEM-F	GCGGAACCCCTATTTG	50	963	26
	TEM-R	ACCAATGCTTAATCAGTGAG			
*SHV*	SHV-F	AGCCGCTTGAGCAAATTAAAC	60	713	27
	SHV-R	ATCCCGCAGATAAATCACCAC			
*CTX group1*	CTX group1-F	TTAGGAARTGTGCCGCTGYA	52	688	27
	CTX group1_2-R	CGATATCGTTGGTGGTRCCAT			
*CTX group2*	CTX group2-F	CGTTAACGGCACGATGAC	52	404	27
	CTX group1_2-R	CGATATCGTTGGTGGTRCCAT			
*CTX group9*	CTX group9-F	TCAAGCCTGCCGATCTGGT	52	561	27
	CTX group9-R	TGATTCTCGCCGCTGAAG			
*CTX group8*	CTX group8-F	AACRCRCAGACGCTCTAC	52	326	27
	CTX group8-R	TCGAGCCGGAASGTGTYAT			
*CTX M-15*	CTX M-15-F	CACACGTGGAATTTAGGGACT	50	995	25
	CTX M-15-R	GCCGTCTAAGGCGATAAACA			
*gyrA*	gyrA-F	AAATCTGCCCGTGTCGTTGGT	58	344	29
	gyrA-R	GCCATACCTACGGCGATACC			
*parC*	parC-F	CTGAATGCCAGCGCCAAATT	57	168	29
	parC-R	GCGAACGATTTCGGATCGTC			
*qnrA*	qnrA-F	ATTTCTCACGCCAGGATTTG	53	516	28
	qnrA-R	GATCGGCAAAGGTTAGGTCA			
*qnrB*	qnrB-F	GATCGTGAAAGCCAGAAAGG	53	469	28
	qnrB-R	ACGATGCCTGGTAGTTGTCC			
*qnrS*	qnrS-F	ACGACATTCGTCAACTGCAA	53	417	28
	qnrS-R	TAAATTGGCACCCTGTAGGC			

Abbreviations: Bp: base pairs, F: forward, R: reverse.

**Table 2 antibiotics-11-01817-t002:** Phenotypic and genotypic characteristics of *Klebsiella pneumoniae* isolates.

Isolate	MLST	Isolation Date	Source	Originating Site	ESBL	β-Lactamases	Resistance (R) or Susceptibility (S) to Antimicrobials									
							CEF	CAZ	FEP	ETP	IMI	GEN	AMK	CIP	STX	NI
S-1226	307	Oct/2018	Wound secretion	A/Surg	+	CTS	R	R	R	S	S	S	—	R	R	—
CC4 ^a^	307	Nov/2018	Urine	B/IM	+	CTS	R	R	R	S	S	R	—	R	R	—
H18-2354	307	Nov/2018	Blood	A/Surg	+	CTS	R	R	R	S	S	R	S	R	R	—
O-3651	307	Nov/2018	Urine	A/ICU	+	CTS	R	R	R	S	S	R	S	R	R	R
HR-0054 ^a^	307	Feb/2019	Rectal swab	A/Out	+	CTS	R	R	R	—	I ^b^	R	I	R	R	—
164605	307	Mar/2019	Endotracheal secretion	C/ICU	+	CTS	R	R	R	S	S	R	S	R	R	I
S-0734 ^a^	152	May/2019	Wound secretion	A/Ort	+	CTS	R	R	R	S	S	R	—	R	R	I
365	18	Jun/2019	Urine	B/IM	−	ND	S	S	S	S	S	S	S	S	S	R
O-2659	29	Nov/2019	Urine	A/IM	−	ND	S	S	S	S	S	S	S	S	S	I
O-2723	405	Nov/2019	Urine	A/Neu	+	CTS	R	R	R	S	S	R	S	I	R	—
CC5	2073	Nov/2019	Urine	B/IM	+	CTS	R	R	R	S	S	R	I	S ^c^	R	R

^a^ Resistant to ampicillin/sulbactam. ^b^ Intermediate resistance to meropenem. ^c^ Resistance to nalidixic acid. Abbreviations: AMK: amikacin; CAZ: ceftazidime; CEF: cephalothin; CIP: ciprofloxacin; CTS: CTX-M-15, TEM, and SHV β-lactamases; ESBL: extended-spectrum β-lactamase; ETP ertapenem; FEP: cefepime; GEN: gentamicin; ICU: intensive care unit; IM: internal medicine ward; IMI: imipenem; MLST: multilocus sequence typing; ND; not determined; Neu: neurology ward; NI: nitrofutantoin; Ort: orthopedics ward; Out: outpatient; Surg: surgical ward; SXT: trimethoprim-sulfamethoxazole.

**Table 3 antibiotics-11-01817-t003:** Distribution by ST of QRDR and PMQR genes in isolates resistant to quinolones.

Isolate	ST	PQMR	QRDR	gyrA		parC	
				83	87	80	87
S-1226	307	*qnrB*	2	**Ile** (**ATC**) *****	Asp (GAC)	**Ile** (**ATT**) *****	Glu (GAA)
CC4	307	*qnrB*	2	**Ile** (**ATC**) *****	Asp (GAC)	**Ile** (**ATT**) *****	Glu (GAA)
H18-2354	307	*qnrB*	2	**Ile** (**ATC**) *****	Asp (GAC)	**Ile** (**ATT**) *****	Glu (GAA)
O-3651	307	*qnrB*	2	**Ile** (**ATC**) *****	Asp (GAC)	**Ile** (**ATT**) *****	Glu (GAA)
HR-0054	307	*qnrB*	2	**Ile** (**ATC**) *****	Asp (GAC)	**Ile** (**ATT**) *****	Glu (GAA)
164605	307	*qnrB*	2	**Ile** (**ATC**) *****	Asp (GAC)	**Ile** (**ATT**) *****	Glu (GAA)
S-0734	152	*qnrB*	3	**Phe** (**TTC**) *****	**Ala** (**GCC**) *****	**Ile** (**ATC**) *****	Glu (GAA)
O-2723	405	*qnrB*	0	Ser (TCC)	Asp (GAC)	Ser (AGC)	Glu (GAA)
CC5	2073	*qnrA*	0	Ser (TCC)	Asp (GAC)	Ser (AGC)	Glu (GAA)

Abbreviations: PMQR: plasmid mediated quinolone resistance; QRDR: quinolone resistance determining region; ST: sequence typing. * Substitutions: Ala: alanine; Asp: aspartic acid; Glu: glutamine; Ile: isoleucine; Phe: phenylalanine; Ser: serine.

**Table 4 antibiotics-11-01817-t004:** Risk factors identified in *Klebsiella pneumoniae* isolates.

Variables	ST307 (*n* = 6)	Other ST (*n* = 5)	*p* Value
Sex			>0.99
Female *n* (%)	2 (33)	2 (40)	
Male *n* (%)	4 (67)	3 (60)	
Age, years, median (IQR)	66 (63, 75)	63 (56, 66)	0.64
Risk factors			
Hospitalized ≥2 d in the prior 90 d	5 (83)	3 (60)	
Antibiotic treatment in the prior 90 d	4 (67)	2(40)	
Wound care at home	2 (33)	1 (20)	
Outpatient chemotherapy	2 (33)	—	
History of immunosuppressive therapy	2 (33)	—	
Hemodialysis in the prior 90 d	—	—	
At least 1 risk factor	6 (100)	3 (60)	0.18
No known risk factors	—	2 (40)	0.06
ESBL carrier	6 (100)	2 (40)	0.06

Abbreviations: d: days; ESBL: extended-spectrum β-lactamase, ST: sequence typing.

## Data Availability

The data presented in this study are available within the article.

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
