# Peer review of "Molecular Genetic Epidemiology of an Emerging Antimicrobial-Resistant *Klebsiella pneumoniae* Clone (ST307) Obtained from Clinical Isolates in Central Panama"

_antibiotics, 2022, doi:10.3390/antibiotics11121817_

Round 1
Reviewer 1 Report
The Authors aimed to discuss the Molecular Genetic Epidemiology of an Emerging Antimicrobial-Resistant Klebsiella pneumoniae Clone (ST307) Obtained from Clinical Isolates in Central Panama. The study carries good information that is promising and interesting from a scientific and practical point of view, but the manuscript requires minor revisions for further processing.
Comments:
1.Your study focused on ESBL genes and quinoline resistance, whether there is a rationale to select these types only.
2.Study should have been advanced by the detection of other antibiotic resistance genes particularly carbapenem and colistin resistance, which will depict a clear picture of the current resistance pattern.
3.The introduction should be enriched with the latest resistance pattern by adding content and references from the below articles. The information can be added and cited below articles:
https://doi.org/10.3389/fcimb.2019.00193
https://journals.asm.org/doi/10.1128/CMR.00064-16
https://doi.org/10.3390/pharmaceutics14020295
4.Entire manuscript requires more language and grammar correction
Author Response
REVIEWER 1
Comments and Suggestions for Authors
The Authors aimed to discuss the Molecular Genetic Epidemiology of an Emerging Antimicrobial-Resistant Klebsiella pneumoniae Clone (ST307) Obtained from Clinical Isolates in Central Panama. The study carries good information that is promising and interesting from a scientific and practical point of view, but the manuscript requires minor revisions for further processing.
Comments:
1.Your study focused on ESBL genes and quinoline resistance, whether there is a rationale to select these types only.
Response: Thanks for your valuable comments. We focused mainly on genes encoding for ESBL and quinolone resistance because the strains tested showed predominantly a resistance phenotype to these two groups of antibiotics (beta-lactams and quinolones).
2.Study should have been advanced by the detection of other antibiotic resistance genes particularly carbapenem and colistin resistance, which will depict a clear picture of the current resistance pattern.
Response: Many thanks for your valuable comment. The study was designed to explore resistance phenotypes and, consequently, their associated genes. No strains with carbapenem or colistin resistance phenotypes were recorded in this study, so we did not study resistance genotypes associated with these types of antibiotics. We focused mainly on the ESBL and quinolone genes because the strains analysed showed predominantly a resistance phenotype to these two groups of antibiotics.
3.The introduction should be enriched with the latest resistance pattern by adding content and references from the below articles. The information can be added and cited below articles:
https://doi.org/10.3389/fcimb.2019.00193
https://journals.asm.org/doi/10.1128/CMR.00064-16
https://doi.org/10.3390/pharmaceutics14020295
Response: Thank you for your comment and suggestions. We have added new content in the introduction and included the suggested associated references.
4.Entire manuscript requires more language and grammar correction
Response: Thank you very much for highlighting the need for further linguistic and grammatical correction. We have already gone through the entire manuscript to make the linguistic corrections.
Reviewer 2 Report
In the world today, increased antimicrobial resistance is a serious problem. Multidrug-resistant strains (MDR) emerge among bacteria, which poses a great threat to the health and life of patients. Kebsiella pneumoniae is one of the species belonging to the Enterobacteriaceae family, responsible for nosocomial and environmental infections.
Although the study shows interesting findings and improve the knowledge in the field, minor revisions are required to improve the readability and the comprehension of the research.
- The paper deals with the topic of actuality.
- The paper is clear, well written, and the organization is very good.
- The references are up to date, and they are well organized according to the format required by the journal.
line 124- what does the author mean by ,, secretions''?
I recommend authors cite the following paper as it contains very interesting information doi: 10.3390/antibiotics10070868. PMID: 34356789; PMCID: PMC8300768; DOI: 10.3390/antibiotics11040503
Author Response
REVIEWER 2
Comments and Suggestions for Authors
In the world today, increased antimicrobial resistance is a serious problem. Multidrug-resistant strains (MDR) emerge among bacteria, which poses a great threat to the health and life of patients. Kebsiella pneumoniae is one of the species belonging to the Enterobacteriaceae family, responsible for nosocomial and environmental infections.
Although the study shows interesting findings and improve the knowledge in the field, minor revisions are required to improve the readability and the comprehension of the research.
- The paper deals with the topic of actuality.
- The paper is clear, well written, and the organization is very good.
- The references are up to date, and they are well organized according to the format required by the journal.
line 124- what does the author mean by ,, secretions''?
Response: Many thanks to the reviewer for valuable comments. We have now clarified that the secretions are one endotracheal and two wound samples.
I recommend authors cite the following paper as it contains very interesting information doi: 10.3390/antibiotics10070868. PMID: 34356789; PMCID: PMC8300768; DOI: 10.3390/antibiotics11040503
Response: Thank you for your comments and suggestions. We have added new content in the introduction and discussion, as suggested by the reviewer. In addition, we have included the associated references.
Reviewer 3 Report
Please make sure that, according to journal rules, all bacterial species names are written in italics (e.g. line 84, 183).
Please add a section in the Discussion about the other beta-lactamases genes analyzed (TEM and SHV), apart from CTX-M-15.
Line 39 - please correct "contributomg" in "contributing"
Line 65 - ", which also carry genes" the subject is lacking, so for example add in the begining of the phrase "carry plasmids harboring CTX-M-15" or end phrase before dot and begin a new phrase "These plasmids also carry genes".
Line 90 - "the term cut-off" is generaly used for linear measurements to separate negative from positive samples, here in antimicrobial testing the term "breakpoint" is usually used.
Line 108 - please replace "PCR products" with "PCR amplicones" or "amplified sequences"
Line 113 - please add genes after ParC
Line 114 - please replace "amplified samples" with "PCR amplicones" or "amplified sequences"
Line 121 - please add in the legend of the tables explanation for F and R primers (forward and reverse?).
Line 124 - please detail the sample type collected, e.g. purulent secretions (also in Table 2).
Table 2 - please rectify the legend, in table you have have asteriscs (*) that are not explained.
Line 169 - please rectify "RAM" abbreviation is actually "AMR"
Line 175 - please replace "health" with "healthcare"
Line 180 - please replace "whose contribution" with "which contributes".
Author Response
REVIEWER 3
Comments and Suggestions for Authors
Please make sure that, according to journal rules, all bacterial species names are written in italics (e.g. line 84, 183).
Response: Thank you very much for your valuable comments. We have now rewritten all bacterial species names in italics.
Please add a section in the Discussion about the other beta-lactamases genes analyzed (TEM and SHV), apart from CTX-M-15.
Response: Thank you for this suggestion. We have now added to the discussion a section on the other beta-lactamase genes analysed (TEM and SHV), apart from CTX-M-15.
Line 39 - please correct "contributomg" in "contributing"
Response: Thank you for highlighting this error. This has been corrected.
Line 65 - ", which also carry genes" the subject is lacking, so for example add in the begining of the phrase "carry plasmids harboring CTX-M-15" or end phrase before dot and begin a new phrase "These plasmids also carry genes".
Response: Thank you for highlighting this error. We have now rewritten the phrase as: "These plasmids also carry genes"
Line 90 - "the term cut-off" is generaly used for linear measurements to separate negative from positive samples, here in antimicrobial testing the term "breakpoint" is usually used.
Response: Thank you for pointing out this error. We have now changed the word “cut-off” to “breakpoint”.
Line 108 - please replace "PCR products" with "PCR amplicones" or "amplified sequences"
Response: Thanks for your valuable comments. We have now changed the phrase “PCR products” to “PCR amplicones”.
Line 113 - please add genes after ParC
Response: We have now added “genes” after ParC
Line 114 - please replace "amplified samples" with "PCR amplicones" or "amplified sequences"
Response: Thanks for your valuable comments. We have now changed the phrase “PCR products” to “PCR amplicones”.
Line 121 - please add in the legend of the tables explanation for F and R primers (forward and reverse?).
Response: We have now added in the legend of the tables explanation for F: forward and R: reverse primers
Line 124 - please detail the sample type collected, e.g. purulent secretions (also in Table 2).
Response: Many thanks to the reviewer for valuable comments. We have now clarified that the secretions are one endotracheal and two wound samples.
Table 2 - please rectify the legend, in table you have have asteriscs (*) that are not explained.
Response: Thank you for pointing out this error. Actually, the asterisks in table 2 are a typing error.
Line 169 - please rectify "RAM" abbreviation is actually "AMR"
Response: Thank you for highlighting this error. We have now replaced “RAM” by “AMR”
Line 175 - please replace "health" with "healthcare"
Response: We have now replaced “health” with “healthcare”
Line 180 - please replace "whose contribution" with "which contributes".
Response: Many thanks to the reviewer for the valuable comment. We have now replaced “whose contribution” with “which contributes”